# Natural History of Anal HPV Infection in Women Treated for Cervical Intraepithelial Neoplasia

**DOI:** 10.3390/cancers15041147

**Published:** 2023-02-10

**Authors:** Marta del Pino, Isabel Matas, Pilar Carrillo, Cristina Martí, Ariel Glickman, Núria Carreras-Dieguez, Lorena Marimon, Adela Saco, Natalia Rakislova, Aureli Torné, Jaume Ordi

**Affiliations:** 1Institute Clinic of Gynecology, Obstetrics, and Neonatology, Hospital Clínic, Universitat de Barcelona, 08036 Barcelona, Spain; 2Institut d’Investigacions Biomèdiques August Pi i Sunyer (IDIBAPS), 08036 Barcelona, Spain; 3ISGlobal, Hospital Clínic, Universitat de Barcelona, 08036 Barcelona, Spain; 4Department of Pathology, Hospital Clínic, University of Barcelona, 08036 Barcelona, Spain

**Keywords:** human papillomavirus, anal intraepithelial neoplasia, high-grade cervical intraepithelial neoplasia, immunocompetent women

## Abstract

**Simple Summary:**

Women with anal HPV infection treated for HSIL/CIN2-3 should be re-tested for anal HPV after treatment. Women treated for HSIL/CIN with persisting anal HPV infection following treatment are at high risk of HSIL/AIN, suggesting that this subset of patients would benefit from anal exploration. Women with anal HPV infection treated for HSIL/CIN might be at higher risk of recurrent HPV cervical infection even after successful treatment.

**Abstract:**

Women with high-grade squamous intraepithelial lesions/cervical intraepithelial neoplasia (HSIL/CIN) are at high risk of anal human papillomavirus HPV infection, and it has also been suggested that self-inoculation of the virus from the anal canal to the cervix could explain HPV recurrence in the cervix after treatment of HSIL/CIN. We aimed to evaluate the bidirectional interactions of HPV infection between these two anatomical sites. We evaluated 68 immunocompetent women undergoing excisional treatment for HSIL/CIN. Immediately before treatment, samples from the anus and the cervix were obtained (baseline anal and cervical HPV status). Cervical HPV clearance after treatment was defined as treatment success. The first follow-up control was scheduled 4–6 months after treatment for cervical and anal samples. High resolution anoscopy (HRA) was performed on patients with persistent anal HPV infections or abnormal anal cytology in the first control. Baseline anal HPV was positive in 42/68 (61.8%) of the women. Anal HPV infection persisted after treatment in 29/68 (42.6%) of the women. One-third of these women (10/29; 34.5%) had HSIL/anal intraepithelial neoplasia (AIN). Among women achieving treatment success, cervical HPV in the first control was positive in 34.6% and 17.6% of the patients with positive and negative baseline anal HPV infection, respectively (*p* = 0.306). In conclusion, patients with persisting anal HPV after HSIL/CIN treatment are at high risk of HSIL/AIN, suggesting that these women would benefit from anal exploration. The study also suggests that women with anal HPV infection treated for HSIL/CIN might be at higher risk of recurrent cervical HPV even after successful treatment.

## 1. Introduction

Human papillomavirus (HPV) has been described as the main etiological factor for the development of premalignant and malignant lesions in the anogenital tract [1]. The focus of HPV prevention strategies has classically been on the uterine cervix, the most frequent site of HPV infection, premalignant lesions and cancer [2]. In contrast, HPV infections involving the anal canal have been neglected for decades. The low prevalence of anal cancer in the general population [3] probably explains the scant interest of physicians and investigators in prevention strategies for this neoplasia. However, anal cancer has become an increasing health problem. It has been estimated that approximately 14,500 cases of anal cancer in women and 12,500 in men occurred in 2008 worldwide [4]. Moreover, in the last few years, the incidence of anal cancer has been rising in some high-income countries, including Australia, Canada, Denmark, France, the Netherlands, the UK and the USA [5]. One of the reasons proposed to explain the rise in anal cancer incidence is the increased prevalence of HPV infection in the lower genital tract.

The first change in this situation started a couple of decades ago, when a significant prevalence of HPV infection and lesions was observed in some risk groups, such as men who have sex with men, people living with HIV, and other immunosuppressed individuals [6,7]. Interestingly, in the last few years isolated studies have shown that women who have an HPV infection involving the lower genital area, especially those with high-grade squamous intraepithelial lesions (HSILs), are also at high risk for HPV-associated lesions and cancer, not only in other areas of the lower genital tract, such as the vagina and the vulva, but also in other anatomical sites such as the anal canal [3]. This high prevalence can be explained not only by sexual behavioral factors (i.e., anal intercourse) [8], but also by the presence of similar epithelium in the anal canal and the uterine cervix [1,7], which could favor the spread of the virus. Thus, several investigators have suggested that women with HPV-associated premalignant or malignant lesions in the genital or perineal area should undergo routine evaluation of the anal canal using high-resolution anoscopy (HRA) [9,10] or anal cytology [11,12]. However, these approaches are not currently included in any clinical guideline. The lack of a clear definition of the high-risk groups within the immunocompetent population that would benefit from anal cancer screening, as well as the lack of agreement about the best strategies for anal cancer screening and management [13,14,15,16], preclude the introduction of anal screening in women with HPV infection of the lower genital tract. However, epidemiological data show a rising prevalence of anal cancer, with a significant disease burden in women [13]. Thus, correct stratification of the specific risks associated with different clinical situations is key to defining adequate guidelines for optimal screening and management [17].

In addition to the risks related to the development of anal precancer and cancer, HPV infection of the anal canal may also be a risk factor for the persistence of HPV in the genital area. The uterine cervix is by far the most frequent site of HPV-associated infections, premalignant lesions (HSIL) and cancer, and women with HSIL/cervical intraepithelial neoplasia (HSIL/CIN) undergo large-loop excision of the transformation zone (LLETZ) to prevent progression to cancer [18]. However, despite achieving complete removal of the cervical lesion, HPV infection persists or recurs in 5–25% of these treated patients, creating a challenging situation that increases the risk of subsequent cervical lesions and cancer [19,20,21,22]. This persistent/recurrent HPV infection involving the uterine cervix is probably a multifactorial phenomenon, in which several factors may contribute [21,22]. Besides HPV persistence in the uterine cervix or new infection upon re-exposure after treatment, it has recently been suggested that cervical autoinoculation of the virus from the anal canal could also explain HPV recurrence in the uterine cervix after treatment [23]. Nevertheless, the possible relation between cervical and anal HPV infection in women with HSIL/CIN has received little attention, especially in HIV-negative women.

Understanding the natural history of HPV infection in the anal and genital areas in women with HSIL/CIN is a crucial first step to elucidate the possible bidirectional interactions of the infection between the different anatomical areas that may contribute to persistence/recurrence of the infection. Proper understanding of these possible interactions would also be key to developing clinical recommendations on who should be screened for anal HPV infection/lesions, and when. This study is an exploratory analysis to evaluate the interactions between cervical and anal HPV infection in immunocompetent women treated for HSIL/CIN. We aimed to examine the prevalence of anal HPV before and after cervical treatment and to estimate the effect that the infection in one area might have on the other anatomical site. 

## 2. Methods

### 2.1. Selection Criteria

All immunocompetent women undergoing a LLETZ procedure due to HSIL/CIN from December 2019 to December 2020 were eligible for the study. According to the clinical protocols and the national guidelines [24], the criteria for LLETZ were: (1) cervical HSIL diagnosed via colposcopy-directed biopsy and/or endocervical curettage, and (2) a repeated cytological HSIL result in at least two Pap smears performed six months apart in patients histologically diagnosed with cervical low-grade SIL (LSIL) or no lesion following exclusion of vaginal HSIL.

During the study period, 68 women underwent a LLETZ procedure at our institution and were eligible for the study. Following the current recommendations of the Ministry of Health of Spain, HPV vaccination was offered to all women undergoing cervical treatment. The 68 women included in the present study accepted vaccination and were referred for free administration of the nine-valent vaccine. Due to the COVID-19 outbreak in this period, the number of women included was significantly lower than expected. 

The study was approved by the ethics committee of the hospital clinic (HCB/2020/0126). All women were informed of the study and signed the informed consent.

### 2.2. Baseline Cervical and Anal HPV Testing

Immediately before the cervical LLETZ procedure, a sample from the anal canal was obtained for HPV testing (baseline anal HPV status) and liquid-based cytology (LBC). The anal sample was collected by a gynecologist using a cytobrush inserted ∼1.5–2.0 cm into the anus and rotated 360° clockwise for one minute. Subsequently, a second sample from the cervix was obtained for HPV testing (baseline cervical HPV status) and LBC. 

### 2.3. LLETZ Procedure and Evaluation of Immediate Post-Treatment HPV Status

After having obtained the cervical and anal samples, the LLETZ procedure was performed using colposcopy vision. A detailed description of the procedure has been previously reported [21,25,26]. Briefly, following the colposcopy examination, we injected 1 mL of 1% mepivacain into each quadrant of the cervix. Loop size was selected according to the size of the excision area. When there was suspicion of endocervical involvement (transformation zone not completely visible), then again under colposcopy control, we performed a second selective endocervical sweep using a smaller loop (top hat). Following excision, we performed selective coagulation of the bleeding areas via ball diathermy.

Subsequently, a cytobrush was used to collect an endocervical sample of the remaining cervix, which was kept in PreservCyt (Hologic Corp., Marlborough, MA, USA) for intraoperative HPV testing and LBC. The intraoperative HPV testing result was considered as evidence of HPV elimination or persistence, as it allows early identification of HPV status after treatment [27]; a negative result was considered as evidence of HPV elimination (and, therefore, successful treatment), whereas a positive result was considered as HPV persistence after treatment.

Finally, endocervical curettage was performed with a Kervokian curette by systematically scraping the whole endocervical surface. 

### 2.4. HPV Testing and Liquid-Based Cytology 

Cervical and anal samples were stored in PreservCyt solution (Hologic, Malborough, MA, USA) for HPV testing and LBC. In all cases, the samples collected were first submitted to HPV testing. The remaining material (if any) was then processed and evaluated using cytology. 

We used the Cobas HPV test (Cobas 4800, Roche Molecular Diagnostics, GmbH, Mannheim, Germany), based on a real-time polymerase chain-reaction (PCR) system for HPV testing. Using this method, 14 high-risk HPV types can be detected, and it also performs specific genotyping for HPV16 and HPV18.

A ThinPrep T2000 slide processor (Hologic, Inc., Marlborough, MA, USA) was used to prepare thin-layer cytology slides stained using the Papanicolaou method. The Bethesda system was used to evaluate cytology slides [28]. 

### 2.5. Post-Treatment Follow-Up Control

Following the current guidelines [24,29], the first follow-up control was scheduled 4 months after treatment in women with positive margins in the surgical specimen and at 6 months in women with negative margins. In this control, new cervical and anal samples were obtained for HPV testing and LBC. 

All patients with a positive cervical HPV testing, or an abnormal result of the cytology in this follow-up control, were scheduled for a new colposcopy with histological sampling, if required. All patients with a positive anal HPV test or an abnormal cytology result were scheduled for HRA with histological sampling, if required. If all tests performed in the first follow-up control were negative, a second follow-up visit was scheduled 12 months after the first control.

### 2.6. High-Resolution Anoscopy (HRA) and Biopsy

HRA was scheduled in all patients with persistent anal HPV infection or abnormal anal cytology in the first follow-up control. The anoscopy was performed by a gynecologist from the colposcopy unit of our center. Before anoscopy, we performed a digital anorectal examination to discard abnormalities suggestive of anal cancer. An Olympus Evis Exera II CV-180 anoscope (Olympus, Barcelona, Spain) was used to perform the HRA. Acetic acid at 5% was used to perform careful anal evaluation, and the anal canal was stained with an iodine-based solution to highlight areas of abnormal epithelium. When suspicious lesions were detected, an HRA-directed biopsy was performed. Women with anal biopsy results showing HSIL were referred for treatment with trichloroacetic acid. 

### 2.7. Histological Processing and Diagnosis 

All biopsy and surgical specimens were submitted for histological assessment. The specimens were fixed in 10% formalin and routinely embedded in paraffin. Four-micron sections were stained with hematoxylin, eosin and p16 immunohistochemistry (IHC). All IHC analyses were performed using the Roche platform with the CINtec Histology Kit (clone E6H4; Roche-Mtm Laboratories, Heidelberg, Germany). Only cases showing continuous “block” staining of the basal and parabasal layers were considered positive for p16 [30,31]. The cervical and anal samples were classified as negative for dysplasia, low-grade (L)SIL, or HSIL [30]. The anatomical area of the lesion and the three grades of abnormality were specified following the World Health Organization terminology [32]: LSIL/cervical intraepithelial neoplasia grade 1 (LSIL/CIN1), HSIL-CIN2 and HSIL-CIN3 for cervical lesions, and LSIL/anal intraepithelial neoplasia (LSIL/AIN1), HSIL/AIN2 or HSIL-AIN3 for anal lesions. Here, p16 positive staining was considered as a mandatory criterion for the histological diagnosis of HSIL/CIN2-3 and HSIL/AIN2-3. 

The margins of the surgical specimens were identified with ink and carefully examined. The margins were considered as negative if no SIL was detected and positive when SIL of any grade was identified. In the latter situation, the site (exocervical/endocervical) of involvement was reported [21,22].

### 2.8. Data Analysis

The SPSS software (version 28.0.0.0, SPSS, Inc., Chicago, IL, USA) was used for data analyses. The sample size was not calculated. However, for obtaining the highest possible sample size, we included all women undergoing excisional treatment in our institution from December 2019 to December 2020.

In the analysis of HPV genotyping, HPV 16 and/or HPV 18, if present, were considered as responsible for the lesions. These cases were included as “HPV16/18 positive”. Coinfections by other high-risk HPV types were disregarded in these patients. Women showing a negative result for HPV16/18 and a positive result for high-risk HPV types other than 16/18 were classified as “HPV other than 16/18”. Thus, correlations for the HPV genotype could only be calculated for HPV16/18, since infection by HPV other than 16/18 would have been underestimated.

Absolute numbers and percentages were used to present categorical variables, which were compared using the Chi-squared or Fisher exact test, while continuous variables were shown as mean and standard deviation (SD). The analysis of variance test was used to compare means.

## 3. Results

Clinical characteristics of the women included in the study and baseline cervical and anal HPV testing.

The mean age of the 68 women included in the study was 42.0 years (SD 9.4). Forty-nine women (72.0%) were treated due to HSIL/CIN2, fifteen (22.1%) due to HSIL/CIN3, and four women (5.9%) due to a repeated cytological result of HSIL with a histological diagnosis of LSIL/CIN1. Fifty-seven women (83.8%) had only cervical lesions, whereas, besides cervical involvement, eleven (16.2%) showed HPV-associated lesions in other areas of the lower genital tract: three had lesions in the vulva, two in the vagina, one had vulvar and vaginal lesions and six showed lesions in the perineal area. 

Baseline anal HPV testing was positive in 61.8% (42/68) of women. Table 1 shows the characteristics of the women included in the study according to their baseline anal HPV status. Anal sex was reported in 59.0% (29/42) and 53.8% (14/26) of the women with positive and negative baseline anal HPV status, respectively (*p* = 0.301). Baseline anal HPV infection was identified in 90.1% (10/11) and 56.1% (32/57) of the women with multicentric and isolated cervical HPV-associated lesions, respectively (*p* = 0.030). Among the 42 women with positive baseline anal HPV testing, 19 (45.2%) had HPV16/18 (17 being HPV16) and 23 (54.8%) had HPV other than 16/18. 

Baseline cervical HPV testing was positive in 95.6% (65/68) of the women. Table 2 shows the concordance of HPV detection and genotypes between the baseline cervical and anal HPV tests.

### 3.1. Immediate Post-Treatment Cervical HPV Status

Treatment success (an immediate post-treatment HPV negative result in the cervix) was achieved in 63.3% (43/68) of the women: 61.9% (26/42) of the women with a positive baseline anal HPV and 65.4% (17/26) of the women with a negative baseline anal HPV status (*p* = 0.772). 

### 3.2. Cervical and Anal HPV Testing and LBC Result at First Follow-Up Control

The mean time to the first follow-up control was 6.3 months (SD 2.0). Table 3 shows the HPV testing and LBC results in the anal canal and uterine cervix in the first control in the women stratified according to their baseline anal HPV status. Anal HPV testing was positive in the first control in 48.5% (33/68) of the women: 65.2% (15/23) of the women showed a positive cervical HPV infection and 40.0% (18/45) of the women showed a negative cervical HPV infection in the first control (*p* = 0.049). 

Cervical HPV testing in the first control was positive in 34.6% (9/26) of the women with treatment success (an immediate post-treatment HPV negative result in the cervix) but with a positive baseline anal HPV status. This percentage was lower (17.6%; 3/17) among the women who achieved treatment success and had a negative baseline anal HPV status, although the differences did not reach statistical significance (*p* = 0.306). The correlations of the anal and cervical HPV genotype of the 12 women who showed treatment success and positive cervical HPV in the first control are shown in Table 4. Among the women with a positive baseline anal HPV, the correlation for at least one HPV genotype between the baseline anal HPV and cervical HPV in the first control was 100%.

### 3.3. High-Resolution Anoscopy and Anal High-Grade Intraepithelial Lesions 

Ten of the 29 (34.5%) women in whom anal HPV persisted after treatment (positive baseline anal HPV status and positive anal HPV in the first follow-up control) showed an HSIL/AIN: six women had HSIL/AIN2 and four had HSIL/AIN3. Table 5 shows the baseline and first follow-up control results of the cervical and anal HPV genotype, cytology and histological lesions of the 10 women who developed HSIL/AIN. None of the 26 women who had a negative baseline anal HPV testing result had a histological anal lesion in the first follow-up control. Among the 42 women with a positive baseline anal HPV test, none of the 13 who became negative for HPV in the anal canal in the first control showed a histological anal lesion. None of the epidemiological characteristics recorded and shown in Table 1 was found to be associated with the development of HSIL/AIN. 

## 4. Discussion

Our study is novel in its approach of looking at the relationship between the natural history and the bidirectional interactions between cervical and anal HPV infection in women treated for cervical HSIL/CIN2-3. The results of this study show that women treated for HSIL/CIN2-3 are not only at high risk of harboring anal HPV infection but also of developing high-grade lesions in the anal canal. Our study also yields information on which patients are at higher risk of HSIL/AIN2-3 and should undergo HRA to detect these lesions, providing some insight on possible screening strategies for anal lesion detection in this subset of women. Finally, our study also shows that women with anal HPV infection are at higher risk of HPV re-infection of the cervix after successful clearance of the cervical infection due to treatment. 

Although it has been suggested that immunocompetent women with HSIL in the genital area (cervix, vagina or vulva) are at high risk of anal HPV infection and disease, very few reports have analyzed anal HPV infection in these patients [7]. In the present series, the prevalence of anal HPV infection in women with cervical HSIL/CIN2-3 was over 60%, and this percentage increased to 90% in women with multicentric HPV-related lesions, i.e., HSIL involving, in addition to the uterine cervix, the vagina and/or the vulva. This high prevalence of anal infection is in keeping with a few previous reports [3,7,33,34]. 

However, none of these former studies have analyzed which patients with HSIL/CIN and anal HPV infection are at higher risk of HSIL/AIN. This is relevant data, as these women are the subset of patients who could benefit from anal cancer screening and anal canal examination via HRA. In the present series, we found that all HSIL/AIN arose in women with persistent anal infection (positive basal anal HPV status and positive anal HPV testing at the first follow-up control). Remarkably, none of the women who were negative in their baseline anal HPV testing, including four women who became positive in the first control, developed HSIL/AIN. These results confirm the low risk of lesion that acute (or short-term) HPV infection represents [35]. Thus, as shown in our study, although the prevalence of anal HPV is high in women with HSIL/CIN, the risk of harboring an underlying premalignant lesion is not the same for all women. Currently, a risk-based approach has been proposed as the most efficient strategy to develop cervical cancer screening. The objective of this risk-based screening is to identify women with HPV infections who are at higher risk of disease [24,36,37], in order to avoid both under- and overdiagnosis and treatment. This new paradigm could also be applied to anal cancer screening. Even within the suggested “high-risk groups” for anal cancer, not all individuals have the same risk of harboring an underlying or developing anal premalignant or malignant lesion [38,39]. Accurate identification of the patients at higher risk would be the cornerstone for establishing who could really benefit from anal cancer screening and HRA examination. The present series suggests that women with anal HPV infection treated for HSIL/CIN2-3 should be re-tested for anal HPV, and if the infection persists after treatment, they would benefit from an anal examination via HRA.

Interestingly, one-third of the women with a positive baseline anal HPV testing result became spontaneously negative for HPV in the anal canal after removal of the cervical lesion through excisional treatment. This finding suggests that the anal HPV testing result could be due to HPV contamination or self-inoculation from the cervical HPV infection that may disappear after removal of the cervical lesion. Indeed, it has been shown that self-inoculation from cervix to anus and from anus to cervix is quite common [23,40]. However, it is unclear if the risk of developing HPV-associated lesions is the same when the HPV is transmitted through self-inoculation within an individual as when the acquisition is through sexual activity. 

Interestingly, in this study anal sex was reported by more than half of the women, including those with a negative baseline anal HPV infection. This prevalence is higher than the prevalence described in previous reports, which suggests potential underreporting of this sexual practice in some studies [23,40]. Despite anal sex having been described as a risk factor for anal HPV infection, it is already accepted that it is not a necessary condition to acquire anal HPV infection [23], which is in keeping with some of the results of the present series. 

Although this study was not designed to evaluate genotype concordance between cervical and anal infection, 70% of the women with a baseline anal HPV16/18 infection also had HPV16/18 in the cervix. A high degree of genotype concordance between the genital area and the anal area has been previously reported [3,40,41]. These results support the hypothesis that a persistent cervical HPV-positive test after treatment may be attributed to self-inoculation [3]. 

Another interesting result from the present study is that the baseline anal HPV status provides information on the risk of persistent cervical HPV infection after treatment. In a previous study, we showed that an HPV-negative result immediately after cervical treatment was early evidence of treatment success in women treated for HSIL/CIN [27]. Interestingly, in this study, the percentage of women with treatment success (immediate cervical HPV clearance after treatment) whose cervical HPV testing in the first control became positive was higher when they had a positive baseline anal HPV status compared with women with negative baseline anal HPV (34.6 vs. 17.6%, respectively), although the differences were not statistically significant. Moreover, the concordance between basal anal HPV and the cervical genotype in the first follow-up control in this group of women was 100%. These results suggest that, besides re-infection or re-activation of HPV, which may occur in all treated women [21,22], self-inoculation from an active anal HPV infection might explain a new positive result for cervical HPV after treatment. Thus, women with an anal HPV infection might have a higher risk of persistent positive cervical HPV testing after treatment, which should be considered for risk stratification of subsequent disease after treatment. 

This study has several strengths. First of all, this is the first study to evaluate the natural history of HPV infection in the cervix and the anal area in women treated for HSIL/CIN and its timeline correlation. Another important strength is that all the women were prospectively recruited, and all cases were very well characterized through cervical and anal HPV testing via genotyping and LBC. Finally, an important novelty of the present series is the knowledge of treatment success (immediate cervical HPV clearance after treatment). The HPV status immediately after LLETZ allows accurate follow-up of the HPV infection before and after cervical treatment and its correlation with anal HPV infection. 

However, there are also some limitations that should be noted. The most important limitation is the small number of patients included in the study, which may have impaired reaching statistical significance in some of the analyses. The number of women included in the study was lower than initially planned due to the COVID-19 outbreak in 2020, which coincided with the scheduled inclusion period of the study. The pandemic stopped the activity of screening programs and management of lesions for several months, making the inclusion of patients slower and more difficult than initially expected. Another possible limitation of the study is that the HPV testing method did not allow extensive HPV genotyping, and thus the concordance between cervical anal HPV genotypes could not be extensively assessed; however, this was not the objective of the present study, and this lack of information would not modify the relevance of the results presented. 

## 5. Conclusions

In conclusion, the present series provides evidence about the importance of knowing HPV status in both the anus and the cervix in women treated for HSIL/CIN. This knowledge allows for stratifying the risk of anal HPV lesions in women with HSIL/CIN and also helps in assessing the risk of HPV persistence after treatment in women with anal HPV infection due to self-inoculation. Larger longitudinal studies are necessary to confirm these results, which would help in the establishment of anal cancer screening algorithms in high-risk women, and also in the risk stratification of women after HSIL/CIN treatment. 

## Figures and Tables

**Table 1 cancers-15-01147-t001:** Epidemiological characteristics at the baseline visit and pathological results of the women according to anal high-risk human papillomavirus (HPV) status at the first visit. Values are presented as mean ± standard deviation or absolute numbers and percentages.

	Positive Baseline Anal HPV (*n* = 42)		Negative Baseline Anal HPV (*n* = 26)		
	*n*	(%)	*n*	(%)	*p* value *
Age	40.4	±9.1	44.5	±9.1	0.076
Smoking habits					0.227
Non-smoker	19	(45.2)	17	(65.4)	
Ex-smoker	1	(2.4)	0	(0.0)	
Smoker	22	(52.4)	9	(34.6)	
Anal sex					0.301
No	13	(31.0)	12	(46.2)	
Yes	29	(69.0)	14	(53.8)	
HPV-associated disease					0.030
Only cervical involvement	32	(76.2)	25	(96.1)	
Multicentric disease	10	(23.8)	1	(3.8)	
Baseline anal cytology					<0.001
Negative	18	(42.9)	23	(88.5)	
LSIL	14	(33.3)	1	(3.8)	
HSIL	6	(14.3)	0	(0.0)	
NA	4	(9.5)	2	(7.7)	
Indication of conization					0.728
Cytological HSIL	1	(2.4)	1	(3.8)	
Histological HSIL/CIN2–3	41	(97.6)	25	(96.2)	

HPV: human papillomavirus, HSIL: high grade squamous intraepithelial lesion; CIN: cervical intraepithelial neoplasia; LSIL: low grade squamous intraepithelial lesion; NA not available. * Fisher exact test.

**Table 2 cancers-15-01147-t002:** Correlation of human papillomavirus (HPV) types between baseline cervical and baseline anal samples. Values are presented as absolute numbers and percentages.

	Baseline Cervical HPV Genotype						
	HPV16/18		HPV Other than 16/18		HPV Negative		
	*n*	(%)	*n*	(%)	*n*	(%)	*p* Value *
Baseline anal HPV genotype							0.475
HPV16/18	13	(68.4)	5	(26.3)	1	(5.3)	
HPV other than 16/18	12	(52.2)	11	(47.8)	0	(0.0)	
HPV negative	15	(57.7)	9	(34.6)	2	(7.7)	

* Fisher exact test.

**Table 3 cancers-15-01147-t003:** Correlation between cervical and anal human papillomavirus (HPV) result and liquid-based cytology in the first follow-up control according to baseline anal HPV status.

	Positive Baseline Anal HPV (*n* = 42)		Negative Baseline Anal HPV (*n* = 26)		
	*n*	(%)	*n*	(%)	*p* Value *
Anal HPV testing at first control					<0.001
Negative	13	(30.9%)	22	(84.6)	
Positive	29	(69.1)	4	(15.4)	
Anal cytology at first control					0.202
Negative	25	(59.5)	21	(80.8)	
LSIL	7	(16.7)	2	(7.7)	
HSIL	4	(9.5)	0	(0.0)	
NA	6	(14.3)	3	(11.5)	
Cervical HPV testing at first control					0.433
Negative	26	(61.9)	19	(73.1)	
Positive	16	(38.1)	7	(26.9)	
Cervical cytology at first control					0.151
Negative	32	(76.2)	20	(76.9)	
LSIL	10	(23.8)	4	(15.4)	
HSIL	0	(0.0)	2	(7.7)	

LSIL: low grade squamous intraepithelial lesion, HSIL: high grade SIL: squamos intraepithelial lesion; NA: not available. * Fisher exact test.

**Table 4 cancers-15-01147-t004:** Baseline and first follow-up control results of cervical and anal human papillomavirus (HPV) and lesions of the 12 women with negative immediate cervical HPV detection and positive cervical HPV in the first control.

		Baseline Results		Surgical Specimen Results (Uterine Cervix)		Follow-Up Status		
	Multicentric disease	Cervical genotype	Anal genotype	Histological diagnosis	Immediate cervical HPV	Cervical genotype	Anal genotype	Anal biopsy
Patients with positive basal anal HPV test								
1	Yes	other	16/18 + other	HSIL/CIN2	Negative	other	16/18 + other	HSIL/AIN3
2	No	16/18 + other	16/18 + other	HSIL/CIN3	Negative	16/18	Negative	No lesion
3	Yes	16/18 + other	16/18 + other	HSIL/CIN2	Negative	16/18	16/18	HSIL/AIN3
4	No	other	other	HSIL/CIN2	Negative	other	Negative	No lesion
5	No	16/18	other	HSIL/CIN2	Negative	other	other	No lesion
6	No	other	other	HSIL/CIN2	Negative	other	other	No lesion
7	No	16/18 + other	other	HSIL/CIN2	Negative	other	other	No lesion
8	No	16/18	16/18 + other	HSIL/CIN2	Negative	16/18 + other	16/18 + other	No lesion
9	No	Negative	16/18 + other	HSIL/CIN2	Negative	other	Negative	No lesion
Patients with negative basal anal HPV test								
10	No	other	Negative	HSIL/CIN2	Negative	other	16/18 + other	No lesion
11	No	16/18	Negative	HSIL/CIN3	Negative	16/18	Negative	No lesion
12	No	other	Negative	HSIL/CIN2	Negative	other	16/18	No lesion

16/18: HPV genotype 16 and/or 18: Other: HPV other than 16 and/or 18; HSIL: high grade squamous intraepithelial lesion; CIN: cervical intraepithelial neoplasia; AIN: anal intraepithelial neoplasia.

**Table 5 cancers-15-01147-t005:** Baseline and first follow-up control results of cervical and anal human papillomavirus (HPV) and lesions of the 10 women who developed high grade squamous intraepithelial neoplasia/anal intraepithelial lesion (HSIL/AIN).

	Baseline Results				Surgical Specimen Results (Uterine Cervix)	Follow-Up Status			
	Multicentric disease	Cervical genotype	Anal genotype	Anal cytology	Histological diagnosis	Cervical genotype	Anal genotype	Anal cytology	Anal biopsy
1	Yes	other	16/18 + other	HSIL	HSIL/CIN2	other	16/18 + other	negative	HSIL/AIN3
2	Yes	16/18 + other	16/18 + other	LSIL	HSIL/CIN3	16/18 + other	16/18	HSIL	HSIL/AIN2
3	Yes	16/18 + other	16/18 + other	LSIL	HSIL/CIN2	16/18	16/18	LSIL	HSIL/AIN3
4	No	16/18	other	LSIL	HSIL/CIN2	Negative	other	LSIL	HSIL/AIN2
5	No	other	16/18 + other	Negative	HSIL/CIN2	Negative	16/18 + other	HSIL	HSIL/AIN2
6	No	other	other	LSIL	HSIL/CIN2	other	other	NA	HSIL/AIN2
7	Yes	other	16/18 + other	LSIL	HSIL/CIN2	Negative	16/18 + other	LSIL	HSIL/AIN2
8	Yes	16/18	other	LSIL	HSIL/CIN2	Negative	other	HSIL	HSIL/AIN3
9	No	16/18 + other	other	Negative	HSIL/CIN2	Negative	16/18 + other	negative	HSIL/AIN3
10	Yes	other	other	HSIL	HSIL/CIN3	Negative	other	HSIL	HSIL/AIN2

16/18: HPV genotype 16 and/or 18: Other: HPV other than 16 and/or 18; HSIL: high grade squamous intraepithelial lesion; CIN: cervical intraepithelial neoplasia; AIN: anal intraepithelial neoplasia; NA not available.

## Data Availability

Data supporting reported results is unavailable due to privacy and ethical restrictions.

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
