# Peer review of "Natural History of Anal HPV Infection in Women Treated for Cervical Intraepithelial Neoplasia"

_cancers, 2023, doi:10.3390/cancers15041147_

Round 1

Reviewer 1 Report

Thank you for your valuable paper

It was really interesting for me to know the autoinoculation theory of HPV virus. 

The number of your patients is not enough for generalizing your result.

However, I am really impressed with your attempt to do  PAP & HPV tests from the anus and the cervix. I know it is really hard to get data like this.

I have comments

1. In the introduction part, write more epidemiologic information about anal cancer

You already well described the relationship between Cervical lesions and anal cancer. However, for readers, general information about anal cancer is helpful for understanding why your study is valuable 

2. Who did the anal test (anal HPV, PAP)

Gynecologist or another specialist?

3. What about anoscopy? Who did it? please specify it 

Author Response

QUESTION 1: In the introduction part, write more epidemiologic information about anal cancer. You already well described the relationship between Cervical lesions and anal cancer. However, for readers, general information about anal cancer is helpful for understanding why your study is valuable.

RESPONSE:

Following the reviewer’s suggestion some more epidemiological data on anal cancer has been added in the Introduction section. (page 2, lines 45-51 of the revised manuscript):

“However, anal cancer has become an increasing health problem. It has been estimated that approximately 14,500 cases of anal cancer in women and 12,500 in men occurred in 2008 worldwide [4]. Moreover, in the last few years, the incidence of anal cancer has been rising in some high-income countries, including Australia, Canada, Denmark, France, the Netherlands, UK and USA [5]. One of the reasons proposed to explain the rise in anal cancer incidence is the increased prevalence of the HPV infection in the lower genital tract.”

Moreover, to support this new information included in the manuscript two additional references have been added:

[4]  F. Islami, J. Ferlay, J. Lortet-Tieulent, F. Bray, A. Jemal, International trends in anal cancer incidence rates, Int. J. Epidemiol. 46 (2017) 924–938. doi:10.1093/ije/dyw276.

[5] F. Bray, J. Ferlay, I. Soerjomataram, R.L. Siegel, L.A. Torre, A. Jemal, Global cancer statistics 2018: GLOBOCAN estimates of incidence and mortality worldwide for 36 cancers in 185 countries, CA. Cancer J. Clin. (2018). doi:10.3322/caac.21492.

QUESTION 2:  Who did the anal test (anal HPV, PAP), Gynecologist or another specialist?

RESPONSE:

Following the reviewer’s recommendation, we have added the information on who performed the procedure in the methods section(page 3, line 120 of the revised manuscript, underlined the added words):

“The anal sample was collected by the gynecologist using a cytobrush inserted 1.5–2.0 cm into the anus and rotated 360â—¦ clockwise for one minute. Subsequently, a second sample from the cervix was obtained for HPV testing (baseline cervical HPV status) and LBC.”

QUESTION 3:  What about anoscopy? Who did it? please specify it.

RESPONSE:

Following the reviewer’s recommendation, we have added the information on who performed the procedure in the methods section(page 4, lines 217-218 of the revised manuscript, underlined the added words):

“HRA was scheduled in all patients with persistent anal HPV infection or abnormal anal cytology in the first follow-up control. The anoscopy was performed by a gynecologist from the colposcopy unit of our center. Prior to the anoscopy, a digital anorectal examination was carried out to rule out abnormalities suggestive of anal cancer”

Reviewer 2 Report

The study is very interesting and the subject of association of HPV annal infection and HPV cervical infection, especially in immunocompetent population is not well enough studied yet.

I suggest adding a few words regarding the HPV vaccination status and acceptance in Spain.

Also, I noticed that the data for the 12 months follow-up were not presented.

Author Response

QUESTION 1: I suggest adding a few words regarding the HPV vaccination status and acceptance in Spain.

RESPONSE:

We agree with the reviewer that vaccination is a fundamental tool in the prevention of HPV-related cancers. Following the reviewer’s suggestion a new sentence has been added in the methods section to clarify the vaccination status in our setting (page 3, lines 109-112 of the revised manuscript, underlined the added paragraph):

“During the study period, 68 women underwent a LLETZ procedure at our institution and were eligible for the study. Following the current recommendations of the Ministry of Health of Spain, HPV vaccination was offered to all women undergoing cervical treatment. The 68 women included in the present study accepted vaccination and were referred for free administration of the nine-valent vaccine. Due to the COVID-19 outbreak in this period, the number of women included was significantly lower than expected. “

QUESTION 2: Also, I noticed that the data for the 12 months follow-up were not presented

RESPONSE:

This study was designed as an exploratory evaluation of the prevalence of HPV infection in the anal canal in women with cervical HSIL aiming at estimating the potential bidirectional consequences of HPV infection of the lower genital tract and the anus and also of the treatment of the cervical lesion on the anal infection. Although we agree with the reviewer that a follow-up study evaluating the longitudinal interactions of HPV infection in the cervix and the anal canal would be extremely relevant, this was out of the scope of our initial project. Nevertheless, as the reviewer suggests, this is now seriously considered for future studies.
